# Genetic Polymorphisms of *IGF1* and *IGF1R* Genes and Their Effects on Growth Traits in Hulun Buir Sheep

**DOI:** 10.3390/genes13040666

**Published:** 2022-04-09

**Authors:** Ning Ding, Dehong Tian, Xue Li, Zhichao Zhang, Fei Tian, Sijia Liu, Buying Han, Dehui Liu, Kai Zhao

**Affiliations:** 1Key Laboratory of Adaptation and Evolution of Plateau Biota, Northwest Institute of Plateau Biology, Chinese Academy of Sciences, Xining 810001, China; dingning@nwipb.cas.cn (N.D.); tiandehong@nwipb.cas.cn (D.T.); lixue@nwipb.cas.cn (X.L.); zczhang@genetics.ac.cn (Z.Z.); tianfei@nwipb.cas.cn (F.T.); liusj@nwipb.cas.cn (S.L.); hanbuying@nwipb.cas.cn (B.H.); liudehui@nwipb.cas.cn (D.L.); 2University of Chinese Academy of Sciences, Beijing 100049, China; 3Qinghai Provincial Key Laboratory of Animal Ecological Genomics, Northwest Institute of Plateau Biology, Chinese Academy of Sciences, Xining 810008, China; 4Hulun Buir State Farm, Hulun Buir 021000, China; 5Hulun Buir Ecological Industry Academy of Technology, Hulun Buir 021000, China

**Keywords:** *IGF1*, *IGF1R*, association analysis, growth traits, haplotype, Chinese indigenous sheep

## Abstract

The identification of candidate genes and genetic variations associated with growth traits is important for sheep breeding. Insulin like growth factor 1 (*IGF1*) and insulin like growth factor 1 receptor (*IGF1R*) are well-accepted candidate genes that affect animal growth and development. The current study attempted to assess the association between *IGF1* and *IGF1R* genetic polymorphisms and growth traits in Hulun Buir sheep. To achieve this goal, we first identified three and ten single nucleotide polymorphisms (SNPs) in exons of *IGF1* and *IGF1R* in Hulun Buir sheep and then constructed six haplotypes of *IGF1R* based on linkage disequilibrium, respectively. Association studies were performed between SNPs and haplotypes of *IGF1* and *IGF1R* with twelve growth traits in a population encompassing 229 Hulun Buir sheep using a general linear model. Our result indicated three SNPs in *IGF1* were significantly associated with four growth traits (*p* < 0.05). In *IGF1R*, three SNPs and two haplotype blocks were significantly associated with twelve growth traits (*p* < 0.05). The combined haplotype H5H5 and H5H6 in *IGF1R* showed the strong association with 12 superior growth traits in Hulun Buir sheep (*p* < 0.05). In conclusion, we identified SNPs and haplotype combinations associated with the growth traits, which provided genetic resources for marker-assisted selection (MAS) in Hulun Buir sheep breeding.

## 1. Introduction

Growth traits are among the most important economic attributes in sheep breeding and are of great concern to breeding experts. Growth traits, including body weight, average daily gain and body size greatly influence meat productivity, which influences production and profitability in the mutton sheep industry [1]. Studies have revealed that many candidate genes are related to growth traits, among which *IGF1* and *IGF1R* genes are well-accepted candidate genes that affect growth and production performance in livestock [2,3]. Insulin-like growth factor 1 (IGF1) is an endocrine growth factor involved in normal growth and development [4,5,6], fetal development and metabolism [7,8]. Insulin-like growth factor 1 receptor (IGF1R) is encoded by the *IGF1R* gene and is a receptor tyrosine kinase that mediates the actions of IGF1 [9,10].

Significant associations were identified between single nucleotide polymorphisms (SNPs) of the two genes and growth performance in diverse farm animals, including cattle [11,12,13], buffaloes [14], pigs [15,16] and goats [17,18,19,20]. In sheep, it has been reported that SNPs of *IGF1* and *IGF1R* are related to meat production and growth [21,22,23]. Using the PCR restriction fragment length polymorphism (PCR-RFLP) method, Grochowska et al. found a highly significant effect of SNPs in the 5' untranslated (5’ UTR) region of *IGF1* on carcass traits and meat compositions in local sheep breeds in Poland Merino sheep [24]. Negahdary et al. found a significant effect of the 5’ UTR region of the *IGFI* gene on birth weight, weaning weight, 6-month weight, average daily gain from birth to weaning and average daily gain from 6 to 9 months in Makooei sheep [25]. A mutation in intron 12 of the *IGF1R* gene was significantly associated with body weight and growth rate in Pomeranian Coarse wool ewes [26]. Later, associations were found between SNP in exon 3 of *IGF1R* and daily gain in the early developmental stage of Colored Polish Merino sheep [23]. The discovery of associations between genetic polymorphisms and growth traits provides useful information for the genetic improvement in sheep breeding. SNPs in the exon of genes are important because they may cause potentially functional variations, which lead to phenotypic changes in livestock. Since most identified SNPs in *IGF1* and *IGF1R* were located in the 5’ flanking regions, we paid particular attention to genetic variations in the exons of the two genes.

Hulun Buir sheep are one of the representative indigenous sheep breeds in northern China, characterized by their high-grade meat quality and outstanding resistance to stress, such as cold and roughage. A lack of advanced breeding methods leads to poor growth performance compared to commercial breeds. To conduct the genetic improvement and breeding in Hulun Buir sheep, several works have been performed to identify genetic variations that were associated with economic traits in Hulun Buir sheep. It has been shown that the somatostatin receptor 1 (*SSTR1*) gene harbors two SNPs that were remarkably associated with growth traits of Hulun Buir sheep [27]. Based on Genome-wide association studies (GWAS), six SNP loci from 526,225 autosomal markers were greatly associated with carcass traits and chest girth [28]. Candidate genes and SNPs have been reported to be associated with fat deposition and fat metabolism [29,30]. However, no systematic investigations have been reported on the association between genetic polymorphism and the early growth traits in Hulun Buir sheep.

To improve the growth performance of Hulun Buir sheep, we investigated the genetic polymorphisms of *IGF1* and *IGF1R* and their associations with twelve growth traits. By scanning exons of *IGF1* and *IGF1R*, we identified thirteen SNPs in the *IGF1* and *IGF1R* genes and two haplotype blocks involving six haplotypes in 229 Hulun Buir sheep. Among these SNPs and haplotype blocks, six SNPs and two haplotype blocks were remarkably associated with growth traits in Hulun Buir sheep. Notably, we identified two combined haplotypes that demonstrated a strong association with twelve greater phenotypic traits. Conclusively, our study provided useful information and laid the foundation for the genetic breeding of Hulun Buir sheep.

## 2. Materials and Methods

### 2.1. Animals and Data Collection

In total, 229 Hulun Buir lambs (male = 106, female = 123), which were born in March 2019 on Hulun Buir sheep farms (Hulun Buir city, Inner Mongolia, China), were investigated. The animals were grazed in identical conditions. The birth weight (BW), weaning weight adjusted at 4-month-old (WW) and body weight at 9-month-old (NBW) were recorded. Meanwhile, average daily gains (ADG) during birth to weaning, weaning to 9-month-old and birth to 9-month-old periods were calculated. Body height (BH), body length (BL) and chest girth (CG) were measured at weaning and at 9 months of age, respectively. Approximately 1 cm^3^ marginal ear tissues were collected and preserved in 95% ethanol. Genomic DNA of Hulun Buir sheep was extracted using a Tiangen DNA extraction kit (Tiangen Biotech Co., Ltd, Beijing, China) and stored at −20 °C for PCR amplification.

### 2.2. SNP Identification and Genotyping

Four and twenty-one pairs of primers were designed for all exons of *IGF1* and *IGF1R* genes based on the published mRNA sequences (Gene ID: 443318, GenBank No. NM_001009774 (*IGF1*), Gene ID: 443515, GenBank No. XM_027957015 (*IGF1R*)), using Primer Premier 5.0 (Premier Biosoft, Palo Alto, Santa Clara, CA, USA), respectively. The primer information is listed in Appendix A. The PCR contained 100 ng template DNA, 10 pM of each primer, 3.5 μL 10 × PCR buffer, 2.5 mM dNTP, 1 U of Taq DNA polymerase (Takara Biotechnology Co., Ltd., Beijing, China) and double-distilled water (ddH_2_O), to make up a volume of 35 μL. PCR was performed in Thermocycler System (ABI 9700, Applied Biosystems, Waltham, MA, USA) with the following reaction procedure: predenaturation at 94 °C for 5 min, followed by 35 cycles at 94 °C for 30 s, 53–60 °C for 30 s and 72 °C for 40 s, with a final extension at 72 °C for 10 min. PCR products were separated by gel electrophoresis (1.5% agarose), purified using magnetic beads (Agencourt AMPure XP, Beckman Coulter, Krefeld, Germany) and sequenced in an Agilent 3730 sequencer (Agilent Technologies, Santa Clara, CA, USA). The sequencing results were aligned to published sheep *IGF1* and *IGF1R* genes using Chromas 2.0 and SeqMan (DNASTAR software, version 7.1) to identify potential SNPs.

### 2.3. Population Genetics of IGF1 and IGF1R Genes

Genotypic and allelic frequencies were estimated with the direct counting method. Hardy–Weinberg equilibrium (HWE), observed heterozygosity (Ho), expected heterozygosity (He) and effective allele numbers (Ne) were analyzed according to the genotype frequencies of SNPs [31]. Cervus (version 3.0) was used to calculate the polymorphic information content (PIC) of each mutation site [32].

### 2.4. Linkage Disequilibrium Analysis and Haplotype Construction

The extent of linkage disequilibrium (LD) between each pair of SNPs in *IGF1* and *IGF1R* was analyzed according to the value of *r*^2^ using Haploview software (version 4.2) [33]. Haplotype blocks with strong LD of SNPs (*r*^2^ > 0.33) were defined based on the confidence intervals methods [34].

### 2.5. Statistical Analyses

SAS software (version 13.0, SAS Institute) was applied for statistical analyses, and the results were expressed as the mean ± SE (standard error). The associations were carried out between the genotypes and individual growth traits using general linear model (GLM):Y_ij_ = μ + G_i_ + S_j_ + G_i_ × S_j_ + ε(1)
where Y_ij_ is a growth trait measured on an individual animal (BW, WW, NBW, ADG, BH, BL and CG); µ is the mean value of overall; G_i_ is the fixed effect of genotypes of the population (i = 3 levels, except rs600896367 of *IGF1* gene and c.244C>T, rs162159917, rs601806812 and rs193644211 of *IGF1R* gene where i = 2 levels); S_j_ is the fixed effect of sex (j = 2 levels); G_i_ × S_j_ is the interaction effect between sex and genotypes; if the difference of interaction effect between the sex and genotypes is not significant, the general linear model should be reduced as:Y_i_ = μ + G_i_ + ε(2)
The association analysis between haplotype combinations and individual growth traits was analyzed by the following GLM:Y_ij_ = μ + H_i_ + S_j_ + Hi × S_j_ + ε(3)
where Y_ij_ is a growth trait measured on an individual animal (BW, WW, NBW, ADG, BH, BL and CG); µ is the mean value of overall; H_i_ is the fixed effect of haplotype combinations of the population (i = 5 levels); S_j_ is the fixed effect of sex (j = 2 levels); H_i_ × S_j_ is the interaction effect between sex and haplotype combinations; if the difference of interaction effects between the sex and haplotype combinations is not significant, the general linear model should be reduced as:Y_i_ = μ + H_i_ + ε(4)
ε is the random error in the above models. Tukey’s test and Bonferroni corrections were performed for multiple pairwise comparisons between genotypes or haplotype combinations based on SNPs. The p value of 0.05 was defined as statistical significance.

## 3. Results

### 3.1. SNP Detection of IGF1 and IGF1R Genes in Hulun Buir Sheep

We detected three SNPs in the *IGF1* gene and ten SNPs in the *IGF1R* gene in 229 Hulun Buir sheep (Figure 1, Figure 2, Table 1). All of the detected SNPs were transition mutations except for SNP13 (transversion mutation) in exon 19 of *IGF1R*. SNP4 in *IGF1R* was a nonsynonymous mutation resulting in a substitution of Cys for Arg in amino acid sequence, and the rest were synonymous mutations. By searching in the dbSNP database of NCBI, we found that SNP4 and SNP8 in *IGF1R* were two novel single-nucleotide mutations in sheep and will be uploaded to the SNP data bank (Table 1).

### 3.2. Population Genetic Analyses

#### 3.2.1. Genotyping, Genotypic and Allelic Frequencies

Among all the SNPs, the wild types were dominant alleles compared with the mutants (Table 2). Genotyping results showed that SNP1 in *IGF1* as well as SNP4, SNP5, SNP9 and SNP12 in *IGF1R* displayed two genotypes: wild-type homozygotes and mutant heterozygotes, and the remaining eight SNPs showed three different genotypes: wild-type homozygotes, mutant heterozygotes and mutant homozygotes (Table 2). In SNP6–8, heterozygotes showed the highest genotype frequencies compared with wild-type and mutant homozygous. In the remaining 10 SNPs, the wild-type homozygotes had the highest genotype frequencies compared with mutant heterozygotes and homozygotes (Table 2).

#### 3.2.2. Genetic Diversity and Hardy–Weinberg Equilibrium

The allelic frequencies of all 13 SNPs obey the HWE law (*p* > 0.05). The Ne values of SNP2 in *IGF1* and SNP6–SNP8 in *IGF1R* were close to 2. The PIC value showed that the five SNP loci (SNP2, SNP3, SNP6–SNP8) exhibited low polymorphism (PIC < 0.25), while the remaining eight SNPs showed moderate polymorphism in the Hulun Buir sheep population (0.25 < PIC < 0.5) (Table 2).

### 3.3. Effects of Genotypes on Growth Traits

Association analysis was performed between genotypes of the SNPs and growth traits on 229 Hulun Buir sheep. The statistical results were listed in Appendix A.

#### 3.3.1. Effects of SNP Genotypes in *IGF1* on Growth Traits

The GA genotype of SNP1 had significantly greater WCG and NBL than the GG genotype (*p* < 0.05). At the SNP2 locus, the higher NCG was observed in TC genotype than that in the CC genotype but not in the TT genotype (*p* < 0.05). The GG and GA genotypes of SNP3 were significantly associated with greater 4–9 ADG than the AA genotype (*p* < 0.05, Figure 3).

#### 3.3.2. Effects of SNP Genotypes in *IGF1R* on Growth Traits

The mutant homozygotes (CC) of SNP6 had significantly longer NBL than those individuals with the TC genotype (*p* < 0.05, Figure 4A). Significant differences (*p* < 0.05) and extremely significant differences (*p* < 0.01) were found between genotypes of the SNP8 locus with the 11 growth traits out of 4–9 ADG (Figure 4B,C). The genotypes containing the wild–type allele had better phenotypic values than mutant homozygotes. At the SNP13 locus, the individuals with the CC genotype had greater NBW, 0–9 ADG, WCG, NBH and NCG than those with the GG genotype (*p* < 0.05); the CC and CG genotypes were associated with significantly longer NBL than the GG genotype (*p* < 0.05, Figure 4D,E). No significant effects were detected among the remaining seven SNP loci and early growth traits of Hulun Buir sheep (*p* > 0.05).

### 3.4. Linkage Disequilibrium and Haplotype Analysis

A strong linkage disequilibrium (*r*^2^ > 0.33) was observed among SNP5, SNP9 and SNP11, and between SNP6 and SNP7, as well as SNP8 and SNP9 loci in the *IGF1R* gene (Figure 5). In particular, SNP6 to SNP9 loci formed two haplotype blocks. The first haplotype block was composed of SNPs 6 and 7, including three common haplotypes. The haplotypes H1 (TC), H2 (CT) and H3 (CC) occurred at frequencies of 0.537, 0.389 and 0.074, respectively, and five haplotype combinations were generated (Table 3). The second haplotype block was composed of SNP8 and SNP9, including three common haplotypes. The haplotypes H4 (CG), H5 (TG) and H6 (CA) occurred at frequencies of 0.321, 0.581 and 0.098, respectively, and generated five haplotype combinations (Table 4). We did not detect the linkage disequilibrium among three SNP loci (*r*^2^ < 0.33) in the *IGF1* gene (Figure 6).

### 3.5. Effects of Haplotype Combinations on Growth Traits

Association analysis was performed between haplotypes in the *IGF1R* gene and growth traits of 229 Hulun Buir sheep populations. The statistical results were shown in Appendix A. The haplotype block 1 was only significantly associated with NBL, in which H1H3 (TCCC) haplotype combination had significantly longer NBL than those individuals with the H2H3 (CTCC) haplotype combination (*p* < 0.05) (Figure 7A). For haplotype block 2, the sheep with H5H6 (TGCA) haplotype combination was significantly heavier than that of the H4H4 (CGCG) haplotype combination of BW (*p* < 0.05). The individuals with the H5H5 (TGTG) and H5H6 (TGCA) haplotype combinations had significantly greater WW, NBW, 0–4 ADG, 4–9 ADG, 0–9 ADG, WBL, WCG, NBH, NBL and NCG than those with the H4H6 (CGCA) haplotype combination (*p* < 0.05). H5H5 (TGTG) and H5H6 (TGCA) with the wild-type allele T were the predominant haplotype combinations in the experimental population (Figure 7B,C). Therefore, haplotype combinations H1H3 (TCCC), H5H5 (TGTG) and H5H6 (TGCA) can be used as candidate markers for better growth traits of Hulun Buir sheep.

## 4. Discussion

The growth of the animal was subject to growth hormone (GH)-IGF1 somatrotropic axis, in which GH acts as a major regulator for development, growth and anabolic processes. IGF1 modulates the biological actions of GH by binding to its receptor (IGF1R) [35]. IGF system includes IGF ligands and their receptors, which influences glycogenesis, glucogenesis and protein synthesis through the regulation of downstream gene expression and signaling pathways [36]. Among IGF ligands and receptors, IGF1 and IGF1R proteins are crucial regulators of cell growth and metabolism [37,38]. Genetic variation may have an impact on the phenotypic characteristics of animals by influencing the expression and function of the genes [39,40]. Therefore, we inferred that the genetic variation in *IGF1* and *IGF1R* may also influence the growth traits of sheep.

In the present study, we discovered genetic polymorphisms of the *IGF1* and *IGF1R* genes and evaluated their effects on growth traits in Hulun Buir sheep. Our results indicated that *IGF1* and *IGF1R* exhibited low to medium genetic diversity, and some of the genetic variations exhibited a significant association with the growth performance in Hulun Buir sheep. This observation provided SNP marker information, which has potential feasibility for MAS in Hulun Buir sheep breeding schemes.

The Hardy–Weinberg equilibrium of all 13 SNPs indicated the absence of artificial selection of Hulun Buir sheep [41]. In the current study, two novel single-nucleotide polymorphisms were identified, including a nonsynonymous mutation of SNP4. A growing body of evidence has shown that the synonymous mutations could influence phenotypic performance by influencing gene expression through the regulation of mRNA stability and protein expression [42,43,44,45]. Maria et al. reported that the synonymous mutation rs159876393 SNP1 of *IGF1* was associated with milk protein and casein contents in Sarda sheep [46]. A synonymous mutation SNP2 (rs159876393) in exon 2 of *IGF1* was associated with variations in carcass traits of New Zealand Romney Sheep, including carcass weight, backfat thickness and the lean meat percentage [47]. Consistent with previous reports on other sheep breeds, we also identified a strong association of SNP1 and SNP2 with growth traits in Hulun Buir sheep, which indicated that SNP1 and SNP2 of the *IGF1* gene might be related to multiple traits in sheep. A remarkable association was found between SNP3 (rs400398060) of the *IGF1* gene and average daily gain from 4–9 months of age (4–9 ADG) in the present study. This mutant site was also detected in Egyptian Barki sheep and was not correlated with growth traits, indicating that its association might be dependent on the genetic backgrounds of sheep breeds [48]. Few studies reported the association between genetic polymorphisms of the *IGF1R* gene and growth traits in sheep. A significant correlation was detected between average daily gain and an SNP of the *IGF1R* gene in local sheep breed in Poland Merino sheep [23]. The present study reported 10 SNPs in the *IGF1R* gene, and SNP6, SNP8 and SNP13 were significantly associated with growth traits in Hulun Buir sheep. In addition, the sheep with homozygous wild genotype TT of SNP8 and CC of SNP13 had superior growth traits than those with homozygous mutant genotypes CC and GG, suggesting that they could serve as the predominant genotypes.

Generally, linked SNP loci are of much concern because of the existence of substantial LD between causal SNPs [49]. Haplotype combinations involving multiple linked SNP loci may provide more precise information than single SNP markers for association analysis [50,51,52]. In this study, the strong LD suggested that these alleles were tightly linked; thus, we carried out an association analysis between the haplotypes and growth traits. The association and multiple comparison analyses demonstrated that the H5H5 (TGTG) haplotype combination with wild-type alleles was the dominant haplotype. This was consistent with the result that the wild-type allele T of SNP8 was related to better growth traits. Additionally, SNP6-SNP9 formed two haplotype blocks, which displayed a remarkably significant effect on growth traits. Based on the results above, we inferred that the four SNPs did not act independently [53], and SNP6 and SNP8 of *IGF1R* may be causal mutations that affect phenotypic traits [54].

## 5. Conclusions

Conclusively, our analysis showed that SNP1, SNP2 and SNP3 of the *IGF1* gene, SNP6, SNP8 and SNP13, as well as haplotype block 1 and haplotype block 2 of the *IGF1R* gene can be used as candidate markers for early growth traits in MAS of Hulun Buir sheep. Further studies will be conducted to investigate the effects of these SNPs on other economic traits in Hulun Buir sheep. The wild-type alleles of SNP8, haplotype combinations H5H5 (TGTG) and H5H6 (TGCA) in the *IGF1R* gene showed superior growth traits during the early stage. Overall, our study provided important genetic variations, which could serve as potential markers for growth trait selection in Hulun Buir sheep.

## Figures and Tables

**Figure 1 genes-13-00666-f001:**
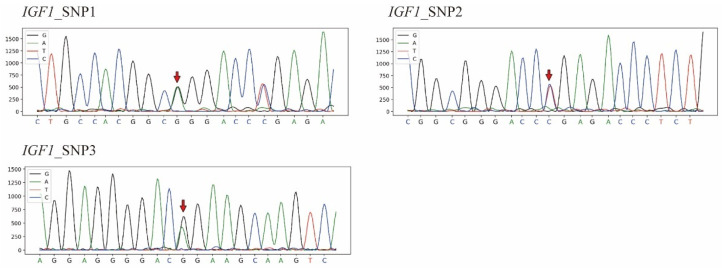
The sequencing peaks for three SNP loci of the *IGF1* gene in Hulun Buir sheep. SNP1: c.144G>A (rs600896367); the arrow indicates the G–A mutation site. SNP2: c.150T>C (rs159876393); the arrow indicates the T–C mutation site. SNP3: c.495G>A (rs400398060); the arrow denotes to the G–A mutation site.

**Figure 2 genes-13-00666-f002:**
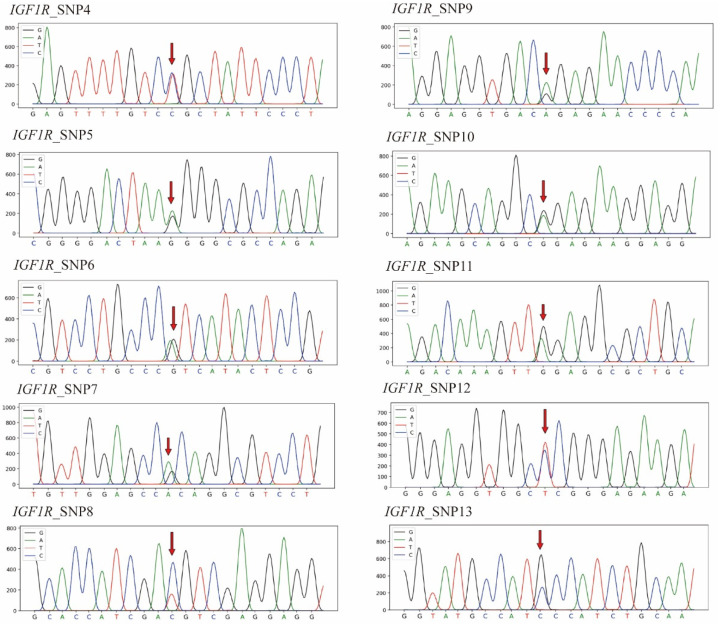
The sequencing peak maps for the ten detected SNP loci of the *IGF1R* gene in Hulun Buir sheep. SNP4: c.244C>T; the arrow denotes to the C–T mutation site. SNP5: c.714G>A (rs162159917); the arrow demonstrates the G–A mutation site. SNP6: c.924T>C (rs161166969); the arrow indicates the T–C mutation site, reverse sequenced as an A–G change. SNP7: c.939C>T (rs162159919); the arrow pinpoints the C–T mutation site, reverse sequenced as a G–A change. SNP8: c.1305T>C; the arrow points to the T–C mutation site. SNP9: c.1320G>A (rs601806812); the arrow indicates the G–A mutation site. SNP10: c.1401A>G (rs161166977); the arrow indicates the A–G mutation site. SNP11: c.1722T>C (rs161166984); the arrow demonstrates the T–C mutation site, reverse sequenced as an A–G change. SNP12: c.2253C>T (rs193644211); the arrow indicates the C–T mutation site. SNP13: c.2634C>G (rs161167008); the arrow pinpoints the C–G mutation site.

**Figure 3 genes-13-00666-f003:**
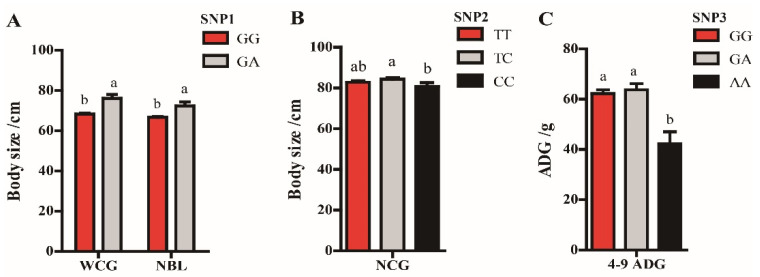
Associations for the SNPs of *IGF1* gene with growth traits in Hulun Buir sheep. (**A**) The comparison of growth traits in SNP1 genotypes of *IGF1* gene; WCG = chest girth at weaning (4-month-old); NBL = body length at 9-months-old. (**B**) The comparison of growth traits in SNP2 genotypes of *IGF1* gene; NCG = chest girth at 9-months-old. (**C**) The comparison of growth traits in SNP3 genotypes of *IGF1* gene; 4–9 ADG = average daily gain from 4 to 9-months-old. Different letters (small letters: *p* < 0.05) above the column indicate significant differences among the different genotypes.

**Figure 4 genes-13-00666-f004:**
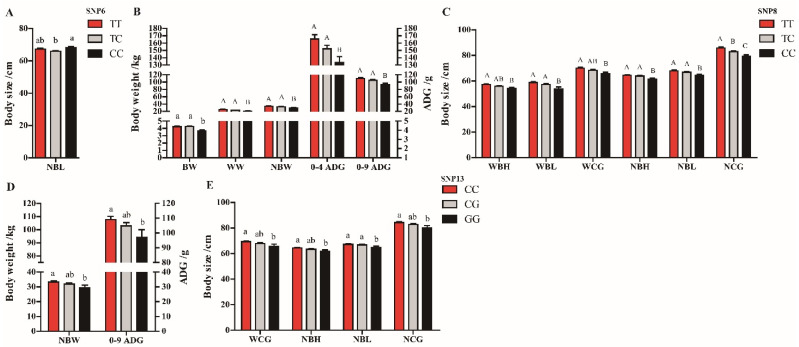
Associations for the SNPs of *IGF1R* gene with growth traits in Hulun Buir sheep. (**A**) Association analysis for different genotypes of SNP6 in the *IGF1R* gene with growth traits; NBL = body length at 9-months-old. (**B**) The comparison of body weight traits in SNP8 genotypes of *IGF1R* gene; BW = birth weight; WW = weaning weight (4-month-old); NBW = body weight at 9-months-old; 0–4 ADG = average daily gain from birth to 4-months-old; 0–9 ADG = average daily gain from birth to 9-months-old. (**C**) Association analyses for different genotypes of SNP8 in *IGF1R* with body size traits; WBH = body height at 4-months-old; WBL = body length at 4-months-old; WCG = chest girth at weaning (4-months-old); NBH = body height at 9-months-old; NBL = body length at 9-months-old; NCG = chest girth at 9-months-old. (**D**) The comparison of body weight traits in SNP13 genotypes of *IGF1R* gene; NBW = body weight at 9-months-old; 0–9 ADG = average daily gain from birth to 9-months-old. (**E**) Association analyses for different genotypes of SNP13 in the *IGF1R* with body size traits; WCG = chest girth at weaning (4-month-old); NBH = body height at 9-months-old; NBL = body length at 9-months-old; NCG = chest girth at 9-months-old. Different letters (small letters: *p* < 0.05; capital letters: *p* < 0.01) above the column indicate significant differences among the different genotypes.

**Figure 5 genes-13-00666-f005:**
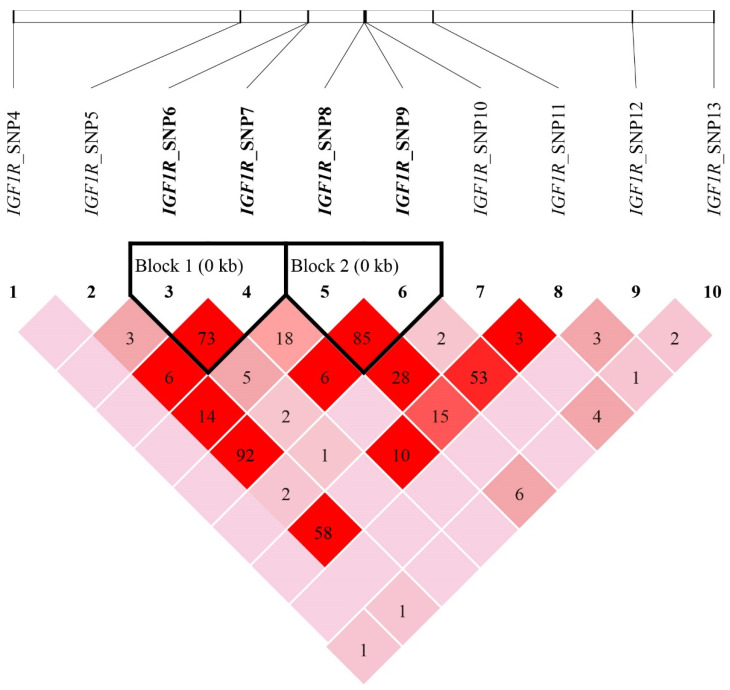
Linkage disequilibrium plot (*r*^2^) and haplotype blocks for SNPs of the *IGF1R* gene in Hulun Buir sheep. The values in boxes are pairwise SNP correlations (*r*^2^); dark red boxes indicate strong LD (*r*^2^ > 0.33) and light red boxes without numbers represent very weak LD (*r*^2^ < 0.001).

**Figure 6 genes-13-00666-f006:**
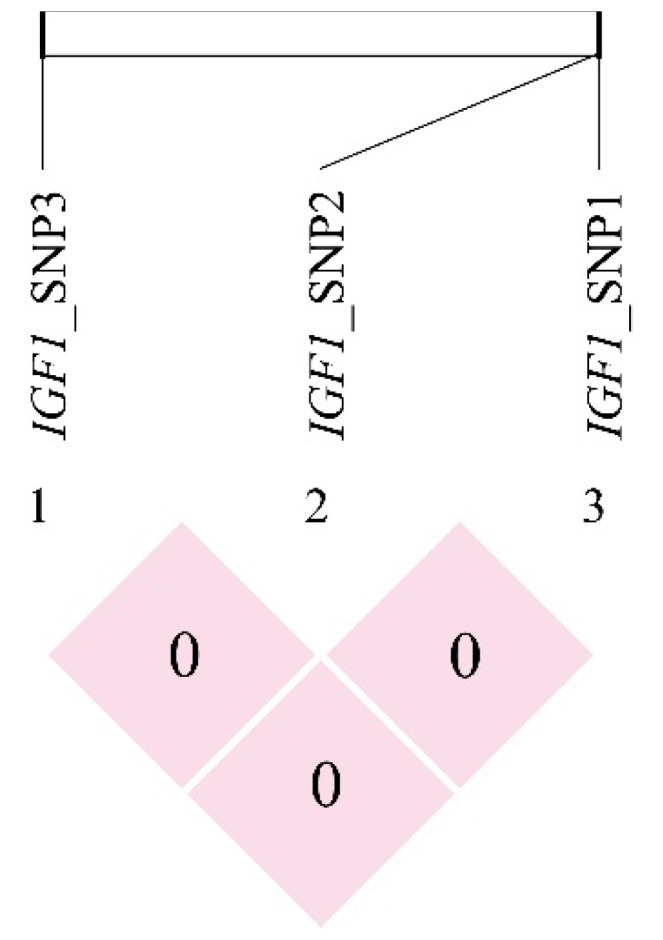
Linkage disequilibrium plot (*r*^2^) and haplotype blocks for SNPs of the *IGF1* gene in Hulun Buir sheep. The values within boxes are pairwise SNP correlations (*r*^2^) and light red boxes represent very weak LD (*r*^2^ < 0.001).

**Figure 7 genes-13-00666-f007:**
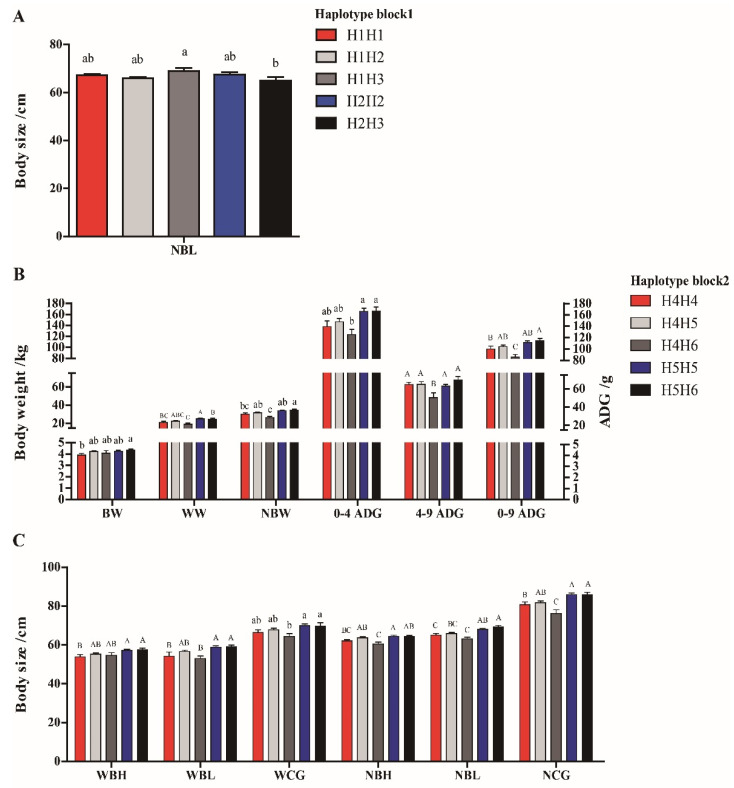
Associations for the haplotype combinations of SNPs in the *IGF1R* gene with growth traits in Hulun Buir sheep. (**A**) Association analysis for the haplotype combinations (block 1) of the *IGF1R* gene with growth traits; NBL = body length at 9–months–old. (**B**) The comparison of body weight traits for the haplotype combinations (block 2) of *IGF1R* gene in Hulun Buir sheep; BW = birth weight; WW = weaning weight (4–month–old); NBW = body weight at 9–months–old; 0–4 ADG = average daily gain from birth to 4–months–old; 4–9 ADG = average daily gain from 4 to 9–months–old; 0–9 ADG = average daily gain from birth to 9–months–old. (**C**) Association analyses for the haplotype combinations (block 2) of *IGF1R* gene with body size traits in Hulun Buir sheep; WBH = body height at 4 months of age; WBL = body length at 4–months–old; WCG = chest girth at weaning (4–month–old); NBH = body height at 9–months–old; NBL = body length at 9–months–old; NCG = chest girth at 9–months–old. Different letters (small letters: *p* < 0.05; capital letters: *p* < 0.01) above the column indicate significant differences among the different haplotype combinations.

**Table 1 genes-13-00666-t001:** The information of SNP in *IGF1* and *IGF1R* in Hulun Buir Sheep.

Gene	Mutant Loci	SNPs	RefSNP	Region	Allele		Amino Acid Variation	Mutation Type
					A	B		
*IGF1*	c.144G>A	SNP1	rs600896367	exon2	G	A	Ala	synonymous
	c.150T>C	SNP2	rs159876393	exon2	T	C	Pro	synonymous
	c.495G>A	SNP3	rs400398060	exon5	G	A	Thr	synonymous
*IGF1R*	c.244C>T	SNP4	-	exon3	C	T	p.Arg81Cys	nonsynonymous
	c.714G>A	SNP5	rs162159917	exon6	G	A	Lys	synonymous
	c.924T>C	SNP6	rs161166969	exon8	T	C	Asp	synonymous
	c.939C>T	SNP7	rs162159919	exon8	C	T	Cys	synonymous
	c.1305T>C	SNP8	-	exon11	T	C	Asp	synonymous
	c.1320G>A	SNP9	rs601806812	exon11	G	A	Thr	synonymous
	c.1401A>G	SNP10	rs161166977	exon11	A	G	Ala	synonymous
	c.1722T>C	SNP11	rs161166984	exon12	T	C	Ser	synonymous
	c.2253C>T	SNP12	rs193644211	exon17	C	T	Ala	synonymous
	c.2634C>G	SNP13	rs161167008	exon19	C	G	Gly	synonymous

**Table 2 genes-13-00666-t002:** Genetic diversity of the SNP loci within *IGF1* and *IGF1R* genes in Hulun Buir sheep population.

Gene	SNPs	Genotype Frequency			Allele Frequency		Ne	Ho	He	PIC	*P* (HWE)
		Wild Type	Hybrid Subtype	Mutant Type	Wild Type	Mutant Type					
		AA	AB	BB	A	B					
*IGF1*	SNP1	0.984	0.016	0	0.992	0.008	1.017	0.016	0.016	0.016	0.057
	SNP2	0.490	0.436	0.074	0.708	0.292	1.705	0.436	0.414	0.328	0.604
	SNP3	0.646	0.329	0.025	0.811	0.189	1.443	0.329	0.307	0.260	0.485
*IGF1R*	SNP4	0.948	0.052	0	0.974	0.026	1.053	0.052	0.051	0.049	0.085
	SNP5	0.810	0.190	0	0.905	0.095	1.208	0.190	0.172	0.157	0.685
	SNP6	0.307	0.451	0.242	0.532	0.468	1.992	0.450	0.498	0.374	0.841
	SNP7	0.368	0.493	0.139	0.615	0.385	1.900	0.494	0.474	0.361	0.818
	SNP8	0.320	0.511	0.169	0.576	0.424	1.955	0.511	0.489	0.369	0.562
	SNP9	0.797	0.203	0	0.898	0.102	1.224	0.203	0.183	0.166	0.085
	SNP10	0.693	0.281	0.026	0.833	0.167	1.385	0.281	0.278	0.239	0.685
	SNP11	0.723	0.247	0.030	0.846	0.154	1.352	0.247	0.260	0.226	0.841
	SNP12	0.931	0.069	0	0.965	0.035	1.072	0.069	0.067	0.065	0.818
	SNP13	0.493	0.416	0.091	0.701	0.299	1.721	0.416	0.419	0.331	0.562

*P* (HWE) = *P* value of Hardy-Weinberg equilibrium, PIC < 0.25 demonstrates low polymorphism, 0.25 < PIC < 0.5 demonstrates medium polymorphism, PIC > 0.5 demonstrates high polymorphism.

**Table 3 genes-13-00666-t003:** Haplotype and haplotype combination analyses of SNPs (block1) in *IGF1R* gene.

Haplotype	SNP6	SNP7	Frequency	Haplotype Combination	Frequency
H1 (TC)	T	C	0.537	H1H1	0.310
H2 (CT)	C	T	0.389	H1H2	0.402
H3 (CC)	C	C	0.074	H1H3	0.096
				H2H2	0.052
				H2H3	0.140

**Table 4 genes-13-00666-t004:** Haplotype and haplotype combination analyses of SNPs (block2) in *IGF1R* gene.

Haplotype	SNP8	SNP9	Frequency	Haplotype Combination	Frequency
H4 (CG)	C	G	0.321	H1H1	0.114
H5 (TG)	T	G	0.581	H1H2	0.367
H6 (CA)	C	A	0.098	H1H3	0.048
				H2H2	0.323
				H2H3	0.148

## Data Availability

Not applicable.

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
