# Peer review of "Genetic Polymorphisms of IGF1 and IGF1R Genes and Their Effects on Growth Traits in Hulun Buir Sheep"

_genes, 2022, doi:10.3390/genes13040666_

Round 1

Reviewer 1 Report

The section 3.5. Effects of Haplotype Combinations on Growth Traits is ambiguous in its present form and should be revised. All content in Table S8 and S9 is marked as statistically significant. However, it is totally unclear how differences among the different genotypes were obtained. The authors should clearly state in the Results/Methods section in which order the haplotypes were compared to each other.

In Tables S2-S9 the significance levels should be marked in classical way (please use signs *, **, and *** for p-values).

The Introduction and Discussion sections highlight the roles of IGF1 and IGF1R genes in growth and results of relevant genotyping in different breeds. However, adding more information on specifics of Hulun Buir sheep, which is Chinese local breed, could improve the relevant parts of the manuscript and will increase the paper readability for international readers.  

Author Response

Response to Reviewer 1 Comments

On behalf of all the contributing authors, I would like to express our sincere appreciations of your letter and reviewer’s constructive comments concerning our manuscript “Genetic Polymorphisms of IGF1 and IGF1R Genes and Their Effects on Growth Traits in Hulun Buir Sheep” (Manuscript ID genes-1642419). These comments are all valuable and helpful for improving our article. According to your and reviewers’ comments, we have made modifications to our manuscript. In the revised version, changes to our manuscript were all highlighted within the documents by using red colored text. Point-by-point responses are listed below this letter.

Point 1: The section 3.5. Effects of Haplotype Combinations on Growth Traits is ambiguous in its present form and should be revised. All content in Table S8 and S9 is marked as statistically significant. However, it is totally unclear how differences among the different genotypes were obtained. The authors should clearly state in the Results/Methods section in which order the haplotypes were compared to each other.

Response 1: Thanks for reviewer’s valuable and helpful comments. It has been revised in the manuscript (Line 257-265).

For haplotype block 2, the sheep with H5H6 (TGCA) haplotype combination was significantly heavier than that of the H4H4 (CGCG) haplotype combination (p < 0.05). The individuals with the H5H5 (TGTG) and H5H6 (TGCA) haplotype combination had extremely significant greater WW, 4-9 ADG, 0-9 ADG, WBL, WCG, NBH, NBL and NCG than those with the H4H6 (CGCA) haplotype combination (p < 0.01); and owned significant better NBW, 0-4 ADG and WCG than the animals with H4H6 (CGCA) haplotype combination (p < 0.05). H5H5 (TGTG) and H5H6 (TGCA) with the wild-type allele T were the predominant haplotype combinations in the experimental population.

The alleles of haplotype combinations has been supplemented in the table S6-S9.

Point 2: In Tables S2-S9 the significance levels should be marked in classical way (please use signs *, **, and *** for p-values).

Response 2: Thanks for reviewer’s valuable and helpful suggestion. We use Tukey’s test and Bonferroni corrections for multiple pairwise comparisons between genotypes or haplotype combinations based on SNPs. Therefor, the significance levels should be marked by the alphabetic method. The statistical methods for association analysis of genotypes or haplotype combinations and growth traits has been explained in the section 2.4 (Line 138-140).

Point 3: The Introduction and Discussion sections highlight the roles of IGF1 and IGF1R genes in growth and results of relevant genotyping in different breeds. However, adding more information on specifics of Hulun Buir sheep, which is Chinese local breed, could improve the relevant parts of the manuscript and will increase the paper readability for international readers.

Response 3: We appreciate the reviewer’s valuable and helpful suggestion. It has been revised in the manuscript (Line 65-74).

To conduct the genetic improvement and breeding in Hulun Buir sheep, several works have been performed to identify genetic variations that were associated with economic traits in Hulun Buir sheep. It has been shown that SSTR1 gene harbors two SNPs that were remarkably associated with growth traits of Hulun Buir sheep [27]. Based on Genome-wide association studies (GWAS), six SNP loci from 526,225 autosomal markers were highly associated with carcass traits and chest girth [28]. Candidate genes and SNPs have been reported to associate with fat deposition and fat metabolism [29, 30]. .However, no systematic investigation have been reported on the association between genetic polymorphism and the early growth traits in Hulun Buir sheep.

Reviewer 2 Report

Standard methodology which allows for repetition of experiment in other populations. Standard statistical methods are used to asnalyse frequerncies of SNPs and effect of genotype/ haplotype on the particular trait.

It is of interest for readers due to clear presentation how research on local population could have an added value to the present state of the art.

With this I have no further commnets nor questions.

Author Response

Response to Reviewer 2 Comments

On behalf of all the contributing authors, I would like to express our sincere appreciations of your letter and reviewer’s high recognition of our manuscript “Genetic Polymorphisms of IGF1 and IGF1R Genes and Their Effects on Growth Traits in Hulun Buir Sheep” (Manuscript ID genes-1642419).

Point : Standard methodology which allows for repetition of experiment in other populations. Standard statistical methods are used to analyse frequerncies of SNPs and effect of genotype/ haplotype on the particular trait. It is of interest for readers due to clear presentation how research on local population could have an added value to the present state of the art. With this I have no further commnets nor questions.

Response : We really appreciate reviewer’s high recognition of the methodology, statistical methods, significance and the positive comments on our works.

Round 2

Reviewer 1 Report

The manuscript was revised properly. 

Author Response

Response to Reviewer 1 Comments

On behalf of all the contributing authors, I would like to express our sincere appreciations of your letter and reviewer’s constructive comments concerning our manuscript “Genetic Polymorphisms of IGF1 and IGF1R Genes and Their Effects on Growth Traits in Hulun Buir Sheep” (Manuscript ID genes-1642419). These comments are all valuable and helpful for improving our article. According to your and reviewers’ comments, we have made modifications to our manuscript. In the revised version, changes to our manuscript were all highlighted within the documents by using red colored text. Point-by-point responses are listed below this letter.

Point 1: The section 3.5. Effects of Haplotype Combinations on Growth Traits is ambiguous in its present form and should be revised. All content in Table S8 and S9 is marked as statistically significant. However, it is totally unclear how differences among the different genotypes were obtained. The authors should clearly state in the Results/Methods section in which order the haplotypes were compared to each other.

Response 1: Thanks for reviewer’s valuable and helpful comments. It has been revised in the manuscript (Line 257-265).

For haplotype block 2, the sheep with H5H6 (TGCA) haplotype combination was significantly heavier than that of the H4H4 (CGCG) haplotype combination of BW (p < 0.05). The individuals with the H5H5 (TGTG) and H5H6 (TGCA) haplotype combination had extremely significant greater WW, 4-9 ADG, 0-9 ADG, WBL, WCG, NBH, NBL and NCG than those with the H4H6 (CGCA) haplotype combination (p < 0.01); and owned significant better NBW, 0-4 ADG and WCG than the animals with H4H6 (CGCA) haplotype combination (p < 0.05). H5H5 (TGTG) and H5H6 (TGCA) with the wild-type allele T were the predominant haplotype combinations in the experimental population.

The alleles of haplotype combinations has been supplemented in the table S6-S9.

Point 2: In Tables S2-S9 the significance levels should be marked in classical way (please use signs *, **, and *** for p-values).

Response 2: Thanks for reviewer’s valuable and helpful suggestion. We use Tukey’s test and Bonferroni corrections for multiple pairwise comparisons between genotypes or haplotype combinations based on SNPs. Therefor, the significance levels should be marked by the alphabetic method. The statistical methods for association analysis of genotypes or haplotype combinations and growth traits has been explained in the section 2.4 (Line 138-140).

Point 3: The Introduction and Discussion sections highlight the roles of IGF1 and IGF1R genes in growth and results of relevant genotyping in different breeds. However, adding more information on specifics of Hulun Buir sheep, which is Chinese local breed, could improve the relevant parts of the manuscript and will increase the paper readability for international readers.

Response 3: We appreciate the reviewer’s valuable and helpful suggestion. It has been revised in the manuscript (Line 65-74).

To conduct the genetic improvement and breeding in Hulun Buir sheep, several works have been performed to identify genetic variations that were associated with economic traits in Hulun Buir sheep. It has been shown that SSTR1 gene harbors two SNPs that were remarkably associated with growth traits of Hulun Buir sheep [27]. Based on Genome-wide association studies (GWAS), six SNP loci from 526,225 autosomal markers were highly associated with carcass traits and chest girth [28]. Candidate genes and SNPs have been reported to associate with fat deposition and fat metabolism [29, 30]. .However, no systematic investigation have been reported on the association between genetic polymorphism and the early growth traits in Hulun Buir sheep.